# Shale oil production and groundwater: What can we learn from produced water data?

**Haoying Wang** *

Department of Business and Technology Management, New Mexico Tech, Socorro, New Mexico, United States of America

* haoying.wang@nmt.edu

## Abstract

As oil production in the Permian Basin surges, the impact of shale production on groundwater resources has become a growing concern. Most existing studies focus on the impact of shale production on shallow freshwater aquifers. There is little understanding of the shale development's impact on other groundwater resources (e.g., deep carbonate aquifers and deep basin meteoric aquifers). The possible natural hydraulic connections between shallow aquifers and formation water suggest such an impact can be consequential. This study explores the relationship between shale production and groundwater using produced water (PW) samples from active unconventional oil wells. Focusing on the most productive portion of the Permian Basin—the four-county region in Southeast New Mexico between 2007 and 2016, a large produced water dataset allows us to analyze the conditional correlations between shale oil production and PW constituents. The results suggest that (1) expanding from primarily conventional wells to unconventional wells during the recent shale boom has led to dramatic increases of the TDS, chloride, sodium, and calcium levels in groundwater (i.e., producing formation). (2) Nearby oil well density positively correlates with the TDS, chloride, and sodium levels in the PW samples.

## Introduction

The scarcity of surface water resources in arid and semi-arid regions has acclimated anthropogenic activities to rely on groundwater to sustain. In some of these regions, economic development has become conflicted with water conservation goals leading to groundwater contamination and over-extraction [1–3]. The Permian Basin in the US desert southwest is one example. The region has experienced rapid growth in shale oil and gas development since the mid-2000s following recent innovations in drilling and fracking technologies. According to a recent report [4], the region accounts for nearly half of the exploration and production activities in the US with over 460 drilling rigs in active operation. Shale fracking injects freshwater mixed with chemical additives and proppants (ceramic or sand) under pressure. During the sequential oil or gas recovery process, produced water (PW) emerges as a byproduct containing mainly formation water (FW) and a small portion of the fracking fluids as flowback [5,6]. On average, over 90% of the PW is naturally occurring FW in the US [7]. The process

**Data Availability Statement:** The water quality and oil production data used in this study are searchable at the following web portal hosted at New Mexico Tech: http://octane.nmt.edu/gotech/Main.aspx. The data have been collected by the Petroleum Recovery Research Center (PRRC) at

New Mexico Tech. They are also the organization that maintains the searchable data web portal at http://octane.nmt.edu/gotech/Main.aspx. If others would like to access the raw data of one of the databases, they should contact the PRRC directly and request the data file. The author has obtained the data in the same way.

**Funding:** The author(s) received no specific funding for this work.

**Competing interests:** The authors have declared that no competing interests exist.

also disturbs groundwater in or near the fractured formation. In the Permian Basin, the PWOR (PW to oil ratio) of shale wells is around three [8]. The extent to which deep formation water being disturbed is often unknown. Hydrogeological research has shown that FW connects to shallow groundwater aquifers in critical ways [3,9–11]. Given the rapidly growing oil and gas production in the Basin (Fig 1), shale production (e.g., fracking and waterflooding), PW disposal, and groundwater quality risk have become both an environmental problem and a policy issue [2,12–15]. Currently, most of the PW is deposited into nonproducing geologic intervals while a small portion gets evaporated or reused (e.g., used in fracking again after some treatment). Some studies suggest that nearly all of the PW in the Southwest have been reinjected into underground reservoirs [2,5,16,17]. PW reinjection may further contaminate FW with unknown long-term consequences because of its magnitude and duration [18,19]. Overall, the impact of shale production on groundwater quality is poorly understood [20].

Shale production affects groundwater through two main pathways: (1) direct disturbance during fracking and recovery processes; (2) reinjection of PW and potential leaks (e.g., due to casing and cement failures). This study explores the correlation between shale oil production and groundwater in the Permian Basin to shed light on the potential impacts. Direct FW samples are difficult to obtain. Existing studies often rely on small sample sizes (e.g., [21]). This study uses a large PW sample taken from producing unconventional wells. The PW samples convey water quality information reflecting the impacts of shale production through both pathways. It is worth noting that even though most of the PW should be tested before reinjecting, the reinjected PW is rarely traced and monitored due to the lack of regulation [22]. Besides, the deposit of PW can go as deep as 10,000 feet (3,048m), which makes it difficult to trace. Fig 2 illustrates the movements of water in the Permian Basin. Our study region is on the west side (i.e., the Delaware Basin), where the PW deposit ranges from 3,000 feet (914m) to over 10,000 feet (3,048m). It is well-known that FW moves over time, so does the reinjected PW [19,23]. Therefore, there exist possibilities that PW and disturbed FW contaminate underground aquifers [3,9]. The oil and gas industry also converts other types of freshwater

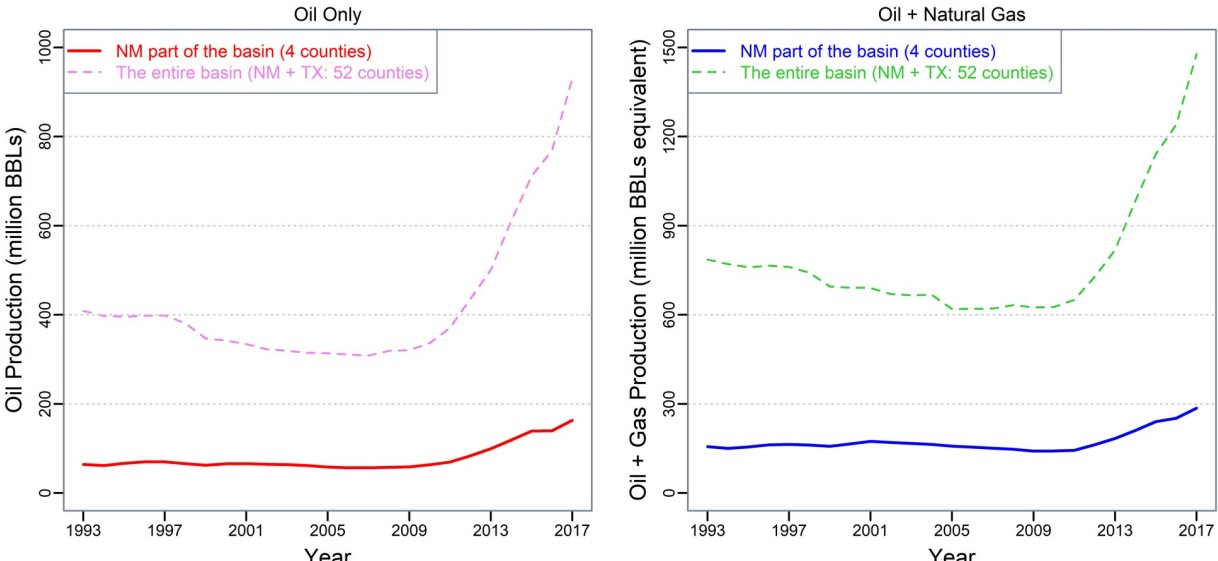

**Fig 1. Oil and natural gas production in the Permian Basin counties (TX + NM).** Note: The author produced the figure in R. Data source: The Petroleum Recovery Research Center at New Mexico Tech and the Railroad Commission of Texas. Natural gas production is converted to oil equivalent based on one MCF natural gas ≈ 0.18 BBL oil (recommended by the US Energy Information Administration).

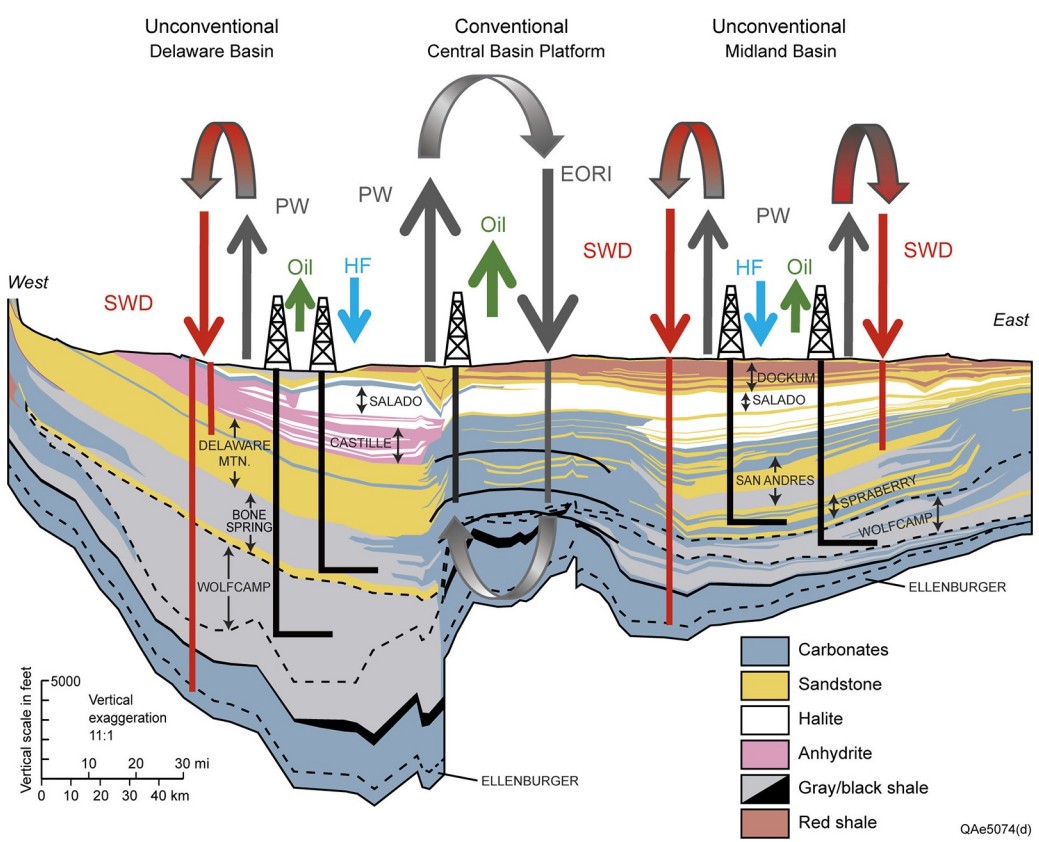

**Fig 2. East-West cross section along the southern margin of the Permian Basin.** Source: This figure is reproduced from [8] with permission from the ACS Publications, the publisher of *Environmental Science & Technology*. Acronyms: SWD—saltwater disposal; PW—produced water; HF—hydraulic fracturing; EORI—enhanced oil recovery injection.

withdrawal into PW. For instance, operators often turn to local farmers and irrigators for water when the water demand for fracking is high. According to industrial sources, oil companies sometimes pay more than $1 per barrel for water used in fracking in the Permian Basin, whereas a local farmer might expect to pay as little as 7.5 cents a barrel [24]. It worsens the water quality issue associated with shale production. In general, we can decompose water quality risk into the probability of contamination and the potential damage. Although the probability of contamination is difficult to quantify and predict, our findings can help infer the potential damage—how has shale FW quality evolved in the context of shale development. This is an important question to answer as exogenous disturbances from shale development may facilitate vertical fluid migration between formation water and freshwater aquifers [21].

Specific to the Permian Basin region, farmers and other water users have faced groundwater quality issues since at least the 1950s, for instance, the increase of chloride content [25]. The concern for water quality issues associated with FW and PW became highlighted in the literature following the Marcellus Shale development in the US [5]. Water quality issues related to shale production have been a challenge to resolve. Among the water-related issues in the Permian Basin, PW reinjection is one of them [8]. PW reinjection reflects a conflict between economic interest (being cost-effective in oil production) and environmental risk (enduring unknown long-term consequences). Many existing studies have examined the impact on

shallow groundwater quality, but studies directly concerning FW and PW are rare. Most of the studies use water samples from private and public water wells to compare quality measures before and after shale development [26–30]. These studies provide valuable observational evidence of the shale production impact on groundwater quality. Still, the literature has yet to establish a full picture of the shale development impact on groundwater quality. The main barriers are the small sample sizes and enormous spatial heterogeneities across geological formations and layers. To the best of our knowledge, [9] is the only peer-reviewed study that links shallow groundwater quality with deep formation water in the context of shale production. As [31] pointed out, our current state of knowledge on shale production and groundwater is still fragmentary understanding. Scientific research, public engagement, and policy implementation that are being outpaced by rapid technological changes need us to be proactive in data collecting and data sharing.

This study aims to explore the potential relationship between shale production and groundwater in the Permian Basin with a relatively large PW sample. Previous studies from the same region have focused on assessing water quality in private wells [30], water budget and the scale of PW [8], and analyzing water demand in shale oil and gas production [2,32]. This study assembles over ten years of PW samples from the New Mexico (NM) portion of the Basin. The data allows us to capture the water profile of the first decade of shale development in the Permian Basin. It also allows us to compare water quality between different stages of shale development. We use a linear regression model to analyze the conditional correlation between oil production and PW constituent levels. The empirical results show that TDS (total dissolved solids), chloride, calcium, and sodium levels had increased substantially after oil production transitioned from primarily conventional wells to unconventional wells. The density of nearby oil wells is positively associated with TDS, chloride, and sodium levels in PW samples drawn from producing unconventional wells. Given that oil and gas production in the Permian Basin expects to continue for another 20–30 years [33], we also discuss potential policy and management implications of the results.

## Data and methods

### Study area and data

This study focuses on the NM portion of the Permian Basin that spreads over four counties: Chaves, Eddy, Lea, and Roosevelt (Fig 3). The area currently contributes over 20% of the total oil and gas production in the basin (see Fig 1). According to a recent report [34], the top two producing counties (Lea and Eddy) accounted for 90% of NM's overall gross receipts tax revenue growth during the 2019 budget year. Other economic impact studies have highlighted the economic importance of these shale development counties [e.g., 35]. Meanwhile, the water table in the area has been declining since at least the 1950s based on historical records. According to [25], the well water depth exceeded 900 feet (274m) in western Eddy County in the 1950s. According to the most recent USGS National Water Information System records, some of the water wells in Eddy County have reached depths of around 2500 feet (762m). The decline of the freshwater table increases the risk of groundwater contamination due to the disturbed formation water from shale development. The region has a population density of 4.757/km$^2$ based on the US Census 2018 estimate. Surface water bodies account for only 0.25% of the total land surface.

Most existing studies concerning the groundwater quality impact of shale production focus on chloride and TDS [e.g., 12,27]. Our data and analysis cover TDS, chloride, sodium, and calcium. The additional constituents can help to better inform the salinity of groundwater in the context of shale production. We derive the data sample from a PW dataset compiled by the

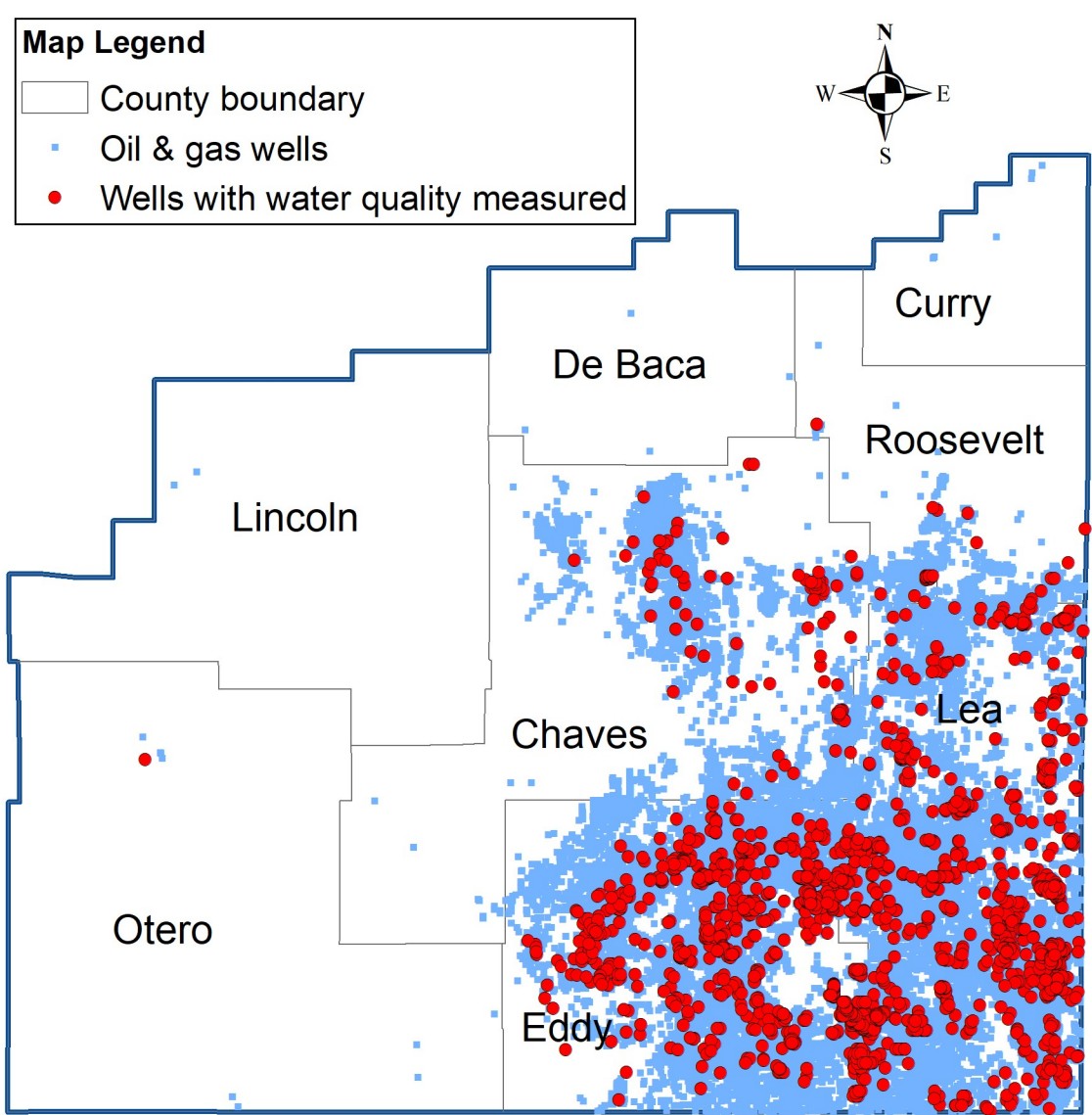

**Fig 3. Study area and data sample locations.** Note: The author produced the figure in ArcGIS 10.4.1. Data source: The Petroleum Recovery Research Center at New Mexico Tech and the US Census (for GIS shapefiles).

Petroleum Recovery Research Center at New Mexico Tech from direct field tests and reports submitted voluntarily by the oil producers. The data collection was part of the New Mexico Water and Infrastructure Data System (NM WAIDS) funded by the US Department of Energy under contract DE-FC26-02NT15134 started in 2002 and recently updated in 2016 [36]. The dataset also contains information on other water constituents (e.g., sulfate, potassium, and bicarbonate), but we excluded them from the analysis due to the small number of observations. Most of the producing wells included in the data sample only reported data occasionally. In the case of one well reporting multiple observations in a given year, we use the average value for the water constituent level. Fig 4 compares the TDS, chloride, sodium, and calcium levels in the PW samples before the shale boom (1993–2001) and after the shale boom (2008–2016). Comparing the median values in the box plots suggests that the levels of TDS, chloride,

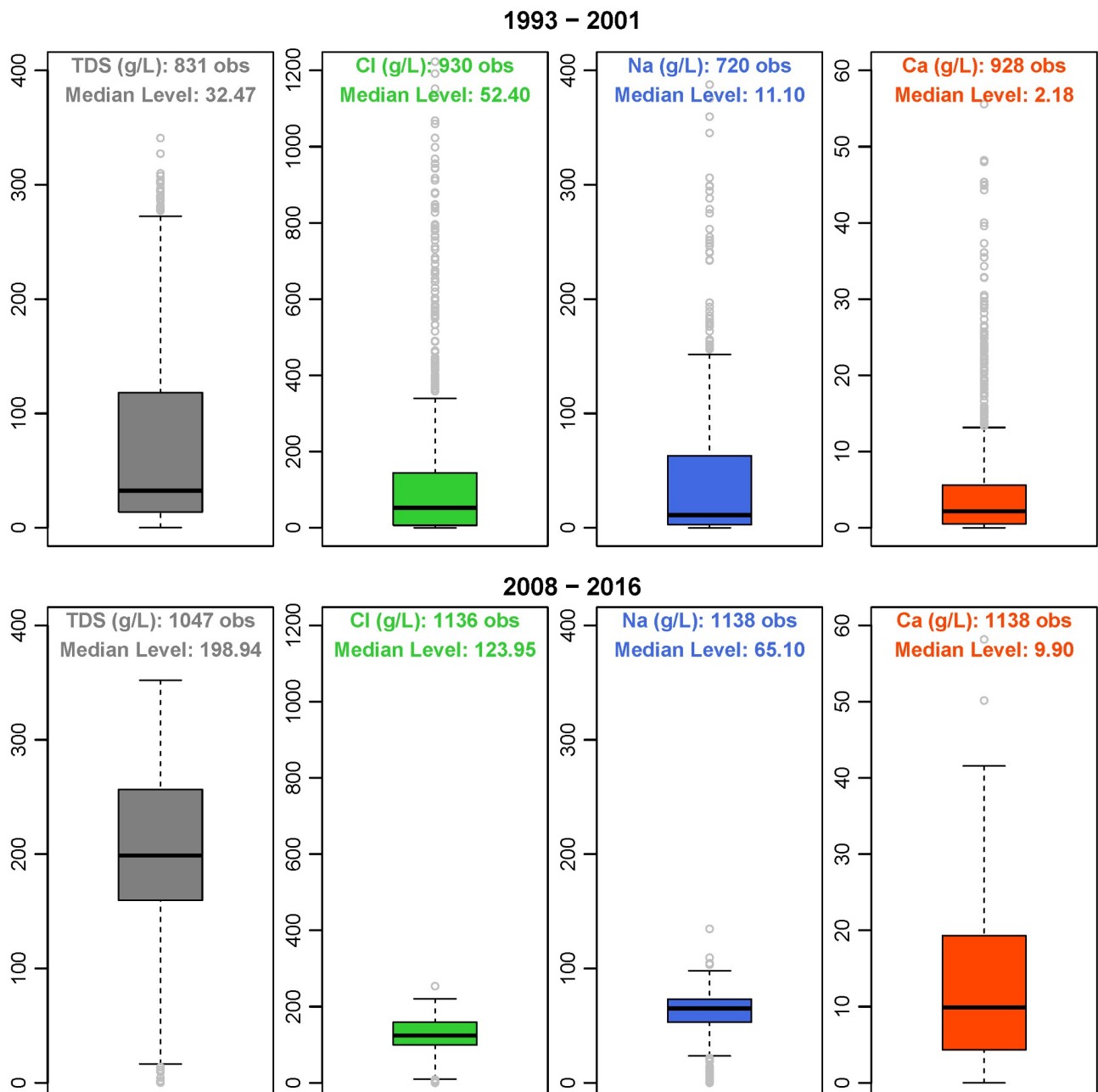

**Fig 4. The levels of TDS, chloride, sodium, and calcium before (1993–2001) and after (2008–2016) the shale boom.** Note: The author produced the figure in R.

sodium, and calcium in PW have substantially increased as a result of shale production. Other than the potential impact of the growing shale production, the changes may also be attributed to the substantial increase of well depth when expanding from primarily conventional wells to unconventional wells. S1 Fig shows the changes of the four constituents from 2007–2011 to 2012–2016 (corresponding to an increase in shale production, see Fig 1; but no substantial changes in well depth), which excludes the PW samples before the shale boom. The levels of all

four constituents had statistically significant increases (statistical tests attached to the figure). It attributes mainly to the growing shale production. Also, we expect that groundwater quality can be affected by other factors such as the age of oil wells and the nature of the geological formation. We further explore these aspects using conditional correlations based on a regression framework focusing on the shale development period: 2007–2016. The well-level oil production data comes from the Petroleum Recovery Research Center at New Mexico Tech as well. The data reports the annual oil production volume of each producing well. We use the data to compute relevant oil production variables (the number of producing wells and the annual oil production within a chosen radius) discussed in conditional correlation sub-section. S1 Table presents the summary statistics of all four constituents and other control variables in the regression analysis.

## Conditional correlation

To explore the relationship between oil production and PW constituent levels, we regress the level of the concerned constituent on a measure of the scale of nearby oil production (*Oil*) controlling other factors including control variables (*Controls*), fixed effects of time-invariant geological features ($\alpha$), and year fixed effects ($\delta$). The coefficient estimates for oil production (*Oil*) give the conditional correlation between oil production and the concerned constituent level. The change of groundwater constituent levels is likely due to both the increase of wells drilled and the growing hydrocarbon production. Hence, we consider two different oil production measures: the total number of nearby producing wells and the annual oil production nearby. The rationale for choosing the linear regression model is to explore the conditional correlation between oil production and PW constituent levels using a relatively large data sample. It is worth noting that we are not trying to identify the deterministic geophysical and geochemical relationship between oil production and groundwater constituent levels. Such a task requires challenging geohydrological field tests and measurements. It also needs dedicated test wells at various depths and locations, which is cost-prohibitive given the current technology. As an alternative, this study relies on existing PW data and an easy-to-implement statistical model to shed some light on the relationship between shale production and groundwater quality. We include all producing oil wells within 2 miles of the tested well. In the Permian Basin, horizontal oil wells can go from 1–2 miles long to beyond 3 miles [4]. 2-mile is a reasonable range to consider when it comes to the potential impact on groundwater. We also exclude all new oil wells (one-year-old or less) to eliminate influences from the flowback of fracking fluids. Mathematically, we set up the following regression model (for TDS, as an example):

$$TDS_{ijt} = \beta_1 Oil_{ijt} + \beta_2 Controls_{ijt} + \alpha_j + \delta_t + \varepsilon_{ijt} \tag{1}$$

where $i$ and $t$ index the location of the tested well and the sample year, respectively. $j$ represents different geological features/formations. The study region consists of five geological features: Delaware Basin, Northwest Shelf, Central Basin Platform, San Simon Channel, and Basin Periphery. $\alpha_j$ are a set of time-invariant dummy variables capturing the unobserved spatial heterogeneities unique to each of the geological features. $\delta_t$ controls any regionwide effects specific to each year (e.g., technological or policy shocks). $\alpha_j$ and $\delta_t$ are commonly referred to as fixed effects in the literature. $\varepsilon_{ijt}$ is the random error. $TDS_{ijt}$ denotes the TDS level in the producing formation—measured by PW from the well (within geological feature $j$) at location $i$ in year $t$. $\beta_1$ is the coefficient of conditional correlation between oil production and PW TDS level. The same regression framework applies to other constituents. The number of valid observations used in each regression is between 500 and 600 (see Table 1 or S1 Table).

**Table 1. Regression of groundwater constituent levels on nearby oil production (sample period: 2007–2016).**

| Constituent Tested | Model Specification | Total number of oil wells | | Total annual oil production | |
|---|---|---|---|---|---|
| | | (1) | (2) | (3) | (4) |
| | Fixed Effects Dummy | None | Geology + Year | None | Geology + Year |
| TDS (mg/L) | Number of oil wells | 334.4126*** (123.2161) | 328.1471*** (115.1689) | | |
| | Oil production (Kbbl) | | | 27.8443 (21.7946) | 18.4814 (17.4281) |
| | Average (oil) well age | 1565.3320*** (537.3866) | 1476.1660*** (506.8118) | 1638.4980*** (542.9227) | 1520.3610*** (511.4846) |
| | Sample size | 524 | | | |
| | Mean TDS (mg/L) | 203,048 | | | |
| | $R^2$ | 0.0296 | 0.1826 | 0.0190 | 0.1710 |
| Chloride (mg/L) | Number of oil wells | 278.6484*** (74.3034) | 252.3123*** (70.5323) | | |
| | Oil production (Kbbl) | | | 12.2649 (13.5544) | 4.3466 (11.2847) |
| | Average (oil) well age | 599.2895* (329.3442) | 374.1464 (316.0641) | 670.0374** (333.6274) | 405.7136 (414.6189) |
| | Sample size | 595 | | | |
| | Mean chloride (mg/L) | 127,363 | | | |
| | $R^2$ | 0.0293 | 0.1565 | 0.0081 | 0.1381 |
| Calcium (mg/L) | Number of oil wells | 33.8454* (18.7696) | 28.6840 (18.4832) | | |
| | Oil production (Kbbl) | | | -0.4744 (2.8158) | -1.5555 (2.5122) |
| | Average (oil) well age | 343.1097*** (75.3113) | 404.3500*** (79.6634) | 347.0260*** (75.7337) | 404.8621*** (93.3568) |
| | Sample size | 558 | | | |
| | Mean calcium (mg/L) | 12,350 | | | |
| | $R^2$ | 0.0425 | 0.1039 | 0.0370 | 0.1004 |
| Sodium (mg/L) | Number of oil wells | 89.1696*** (34.6070) | 69.8458** (33.0467) | | |
| | Oil production (Kbbl) | | | 12.0165** (5.6432) | 7.3764 (5.4058) |
| | Average (oil) well age | 42.6098 (159.1004) | 94.5060 (154.6253) | 85.7026 (159.7392) | 125.6939 (132.5198) |
| | Sample size | 561 | | | |
| | Mean sodium (mg/L) | 62,055 | | | |
| | $R^2$ | 0.0120 | 0.1360 | 0.0083 | 0.1319 |

Note: (1) Oil producing wells within a 2-mile radius (3.22 km) of the tested well are included in the analysis. (2) The heteroscedasticity-consistent standard error of each regression estimate is reported in the parentheses. Throughout the paper, asterisks (*, **, ***) indicate statistical significance at 10%, 5%, and 1% level, respectively, unless otherwise noted.

## Results

All four constituents examined in this study naturally occur in groundwater. Exogenous disturbances can affect their levels of concentration. Table 1 presents the regression results for each constituent (by row panel) across different specifications (by column) between 2007 and 2016 (sample period). In specifications (1) and (2), the *Oil* variable is the total number of producing oil wells within 2 miles. In specifications (3) and (4), the *Oil* variable is the annual oil production within 2 miles. For the control variables, specifications (1) and (3) do not control fixed effects specific to each of the five geological features and each year; specifications (2) and (4) do. The average age of nearby oil wells is a control variable in all specifications. The $R^2$ goodness-of-fit statistics suggest that controlling for geological features and year fixed effects substantially improves the model fit. A variance decomposition using $R^2$ [37] shows that the geological feature fixed effects explain 30~50% of the model fit. Based on the preferred fixed effects specification, the number of oil wells variable on average explains 8% of the model fit across different constituents. The mean well age variable has high explanatory power for

Calcium (45%) but explains very little for others (on average 3.4%). The annual oil production variable has the lowest explanatory power (on average 1.5% across four constituents). It is also worth noting that here the estimated are cross-sectional regression models. The $R^2$ statistics tend to be lower than in a panel data fixed effects model estimated using repeated samples of the same location over time. Moreover, we use the regression models to explore the conditional correlation between PW constituent levels and oil production rather than identifying any causal effects.

The interpretation of results focuses on fixed-effects specifications (2) and (4). When using the number of oil wells to represent the oil production scale (column 2), more oil wells within the 2-mile radius are associated with a higher TDS level. The number of oil wells measures the intensity of nearby drilling & oil production. One additional oil well is associated with an increase in the TDS level by 328.15 mg/L. It is a relatively small impact compared to the effect of well age (1476.17 mg/L per year). The older a shale well, the higher the PW TDS level. Both estimates are statistically significant (*p*-values are 0.005 and 0.004, respectively). They suggest that there is a strong positive correlation between PW TDS level and shale development. We see similar correlations between shale production and PW chloride & sodium levels. The result does not hold when the key independent variable becomes annual oil production within the 2-mile radius. Well depth is another important control variable. However, the raw data has over 50% observations with missing well depth. It was mainly due to the difficulty in determining the formation or layer that the PW sample originated. To include well depth as a control variable, we spatially approximated the missing well depth using average well depth from nearby oil wells. The searching neighborhood size goes from 400 feet (123m) to several miles. Because samples with missing well depth more likely came from deeper oil wells, the spatial interpolation procedure tends to underestimate well depth. Here we use well depth as a control variable only. Its statistical interpretation is not essential. Therefore, a systematic underestimation of the variable is not a concern. S2 Table reports the results with well depth as an additional control variable. The results are qualitatively consistent with what is in Table 1. There are strong positive correlations between the density of nearby oil wells and groundwater constituent levels except for calcium. It is worth noting that controlling well depth improves the model fit substantially. It is consistent with the expectation that groundwater (formation water) gets saltier as the well goes deep.

Switch to specification (4), where the annual oil production within the 2-mile radius represents the oil production scale. The results are statistically insignificant no matter which control variables to include (Table 1 and S2 Table; *p*-values > 0.100). It confirms that drilling density is a more relevant factor for assessing the groundwater impact of shale development. To put the results into context, we use standard deviations. Combining the estimates in Table 1 (column 2) and the summary statistics in S1 Table, a one-standard-deviation increase in the number of nearby oil wells is associated with a 7,606.45 mg/L increase of TDS level (or 3.75% given the mean TDS level at 203,048 mg/L; *p*-value = 0.005). A one-standard-deviation increase in the number of nearby oil wells is associated with a chloride level increase of 5,682.07 mg/L (or 4.46% given the mean chloride level at 127,363 mg/L; *p*-value = 0.000) and a sodium level increase of 1,515.65 mg/L (or 2.44% given the mean sodium level at 62,055 mg/L; *p*-value = 0.035). To sum up, there is a strong positive statistical correlation between shale production and the levels of TDS, chloride, and sodium in PW even after controlling for well age, well depth, geological features, and temporal effects. One should interpret the results in the context of the high level of drilling activities and the rapidly growing oil production in the Permian Basin [35]. It implies that the shale development impact on groundwater may accelerate and bring greater geological and environmental uncertainties.

## Discussion and conclusions

Given the analysis results, TDS, chloride, and sodium are the concerned PW constituents. The implications of their positive correlations with shale production can be significant because of the existing natural hydraulic connections between shallow aquifers and deep formations [9]. These are common constituents naturally occurring in the groundwater. What is more concerning is the chemicals that are not naturally occurring in the groundwater. Many chemicals added for fracking are not currently under regulation by the US Safe Drinking Water Act [5]. For example, glutaraldehyde often needs to be added to fracking fluids to suppress bacterial growth. Its presence in drinking water can irritate tissues and sensitive skin [38]. Even though this study does not directly assess non-naturally occurring constituents in groundwater, its findings reveal the importance of monitoring and regulating these exogenously introduced chemicals during shale production processes. Under certain conditions, the naturally occurring chemicals and exogenously introduced chemicals can react and produce new harmful compounds [39]. In addition, groundwater aquifers with TDS and salinity levels higher than the current use standards may become acceptable for beneficial uses (e.g., irrigation) as technology advances in the future [40]. Hence, it is critical to understand the potential contamination risk of shale production on such aquifers.

One way to mitigate the negative impacts of shale oil production on groundwater quality is by reducing direct PW reinjection. The current high percentage of reinjection leaves a lot of environmental risks in the long term. Nevertheless, underground reinjection is presently the most economical way of PW disposal in the Permian Basin. For small operators, evaporation ponds and trucking to a treatment facility may be more economical. But those practices often lead to other environmental concerns [41–43]. The most environmentally feasible and responsible method of reducing PW reinjection is to reuse the water. The economic feasibility of reusing PW hinges on the cost of treatment technologies (e.g., membrane technology) at the industrial scale [44]. The policy implication is that there is a need to support and develop effective and economical technical solutions for the separation and reuse of dissolved salts from PW and the treatment for naturally occurring toxic chemicals [5]. Any new solutions should allow the continued development of shale resources but in an environmentally justified way. These solutions can be institutional or technological innovations. For instance, The state of NM enacted the Produced Water Act in 2019 to address the environmental and economic issues associated with the PW in the oil and gas industry. Local initiatives like this can help remove institutional barriers around PW reuse while stimulating technological and market innovations [45].

There are two data-related caveats to this study. The discussion here intends to facilitate proper interpretations of the empirical results while leaving some hints for future data collection and research. First, our regression model cannot include well-level fixed effects to better control for unobserved spatial heterogeneities. In general, collecting formation water samples (i.e., brine samples) is difficult and expensive. Many wells in our data sample were only tested once in recent years (2013–2016). Also, the current regulations do not require operators to report groundwater quality regularly [2]. For more accurate and reliable analyses, brine samples from the same well (or formation) should be collected regularly over time. Another limitation is that, as mentioned when discussing the results presented in S2 Table, many PW samples have missing well depth. The missing well depth values are largely due to technical reasons. It is difficult to determine from which layer of the formation the PW sample originated in many cases. Nevertheless, it is often reasonable to assume that the PW sample comes from below the freshwater aquifer if the well depth value is missing.

Overall, using PW samples drawn from shale oil wells, this study explores the relationship between shale oil production and groundwater constituent levels in the Permian Basin. The contribution of this study is two-fold. First, data limitation prevents researchers from directly studying the deterministic geophysical and geochemical relationship between oil production and groundwater constituent levels. This study proposes an easy-to-implement alternative relying on existing PW data and a fixed-effects linear regression model to shed light on the relationship. Second, we show that expanding from primarily conventional wells to unconventional wells during the recent shale boom has led to dramatic increases in the levels of TDS, chloride, sodium, and calcium in PW samples. It attributes mainly to the increases in well depth and the resulted change of geological conditions. Since the onset of the shale development in the mid-2000s, nearby drilling and oil production are found positively correlated with the levels of TDS, chloride, and sodium in PW samples. The correlations are concerning due to the possible natural hydraulic connections between shallow aquifers and formation water. It suggests that shale development may have significant indirect impacts on underground freshwater resources. The findings carry important implications for future research and policymaking regarding PW and shale development. For future research, more data on FW and PW are necessary for improving our understanding of the linkage between freshwater aquifers and formation water. In that sense, this research complements field studies that explore the impact mechanisms between different layers of groundwater in the context of shale production. It is necessary to clarify that this study is positioned as an exploratory study to inspire future research. Currently, it is cost-prohibitive to drill test wells at various locations and depths to study the geophysical and geochemical mechanisms between shale production and groundwater aquifers. As research techniques and data collection evolve, it could become more feasible. When it comes to policymaking, filling in the regulation vacuum around PW in the oil and gas industry is urgent. Both PW and its disposal should be regularly monitored and managed while incentivizing its reuse (within the oil & gas industry and cross-industry).

## Supporting information

**S1 Fig. The levels of TDS, chloride, sodium, and calcium between different stages of shale development (2007–2011 v.s. 2012–2016).** Note: The author produced the figure in R. The t-tests for the difference in means suggest that the increase in the level of concentration is statistically significant for all four constituents. The null hypothesis of no difference is rejected at the 95% confidence level in all cases (p values are 0.0012, 0.0001, 0.0513, 0.0062, respectively). (DOCX)

**S1 Table. Summary statistics of variables (sample period: 2007–2016).** (DOCX)

**S2 Table. Regression of PW constituent levels on nearby oil production controlling well depth.** (DOCX)

## Acknowledgments

The author would like to thank Ms. Martha Cather of the Petroleum Recovery Research Center at New Mexico Tech for her assistance in accessing the data. The author would also like to thank the ACS Publications for allowing this article to reuse Fig 2 from [8]. Please note that further permissions related to the material excerpted should go to the ACS Publications directly.

## Author Contributions

**Conceptualization:** Haoying Wang.

**Formal analysis:** Haoying Wang.

**Investigation:** Haoying Wang.

**Methodology:** Haoying Wang.

**Visualization:** Haoying Wang.

**Writing – original draft:** Haoying Wang.

**Writing – review & editing:** Haoying Wang.

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
