## [Decision Letter · Decision Letter 0]

9 Feb 2021

PONE-D-21-02697

Shale oil production and groundwater: What can we learn from large produced water datasets?

PLOS ONE

Dear Dr. Wang,

Thank you for submitting your manuscript to PLOS ONE. After careful consideration, we feel that it has merit but does not fully meet PLOS ONE’s publication criteria as it currently stands. Therefore, we invite you to submit a revised version of the manuscript that addresses the points raised during the review process.

We look forward to receiving your revised manuscript.

Kind regards,

Jianguo Wang, PhD

Academic Editor

PLOS ONE

Journal Requirements:

2) Please include captions for your Supporting Information files at the end of your manuscript, and update any in-text citations to match accordingly. Please see our Supporting Information guidelines for more information: http://journals.plos.org/plosone/s/supporting-information.

Reviewers' comments:

Reviewer's Responses to Questions

**Comments to the Author**

1. Is the manuscript technically sound, and do the data support the conclusions?

Reviewer #1: Partly

Reviewer #2: Yes

2. Has the statistical analysis been performed appropriately and rigorously? 

Reviewer #1: No

Reviewer #2: Yes

3. Have the authors made all data underlying the findings in their manuscript fully available?

Reviewer #1: Yes

Reviewer #2: Yes

4. Is the manuscript presented in an intelligible fashion and written in standard English?

Reviewer #1: Yes

Reviewer #2: Yes

5. Review Comments to the Author

Reviewer #1: The author used the large data sample from the most productive portion of the Permian Basin – the four-county region in Southeast New Mexico between 2007 and 2016, to analyze the conditional correlations between shale oil production and produced water (PW) constituents. At the same time, he suggested that both produced water and its disposal should be regularly monitored and managed while incentivizing its reuse. Some specific comments are addressed in the following section.

Specific Comments:

1. The innovation of this article is not obvious enough. Please add three highlights before the abstract.

2. The process of processing oil production and groundwater constituent levels data, is not described clearly in Section 2.

3. The selection of the regression model lacks theoretic support. For example, in Eq.(1), it is assumed that there is a linear relationship between parameters , Controls, and . However, in practice, this may not be appropriate. Please clarify it.

4. In this study, all the data were used for training without dividing the test set and the cross-validation set, and whether the model had overfitting problems was not discussed. Therefore, I am negative about the applicability and reliability of the regression model.

5. In Table 1, we see that the levels of TDS, Chloride, Calcium and Sodium correlate with the number of oil wells, oil production and average (oil) well age. Which of these three factors is dominant? I suggest that the author conduct a sensitivity analysis of these factors.

6. The discussion section is disappointing. The author only told us that since the onset of the shale development in the mid-2000s, nearby drilling and oil production were found positively correlated with the levels of TDS, chloride, and sodium in PW samples. However, readers are interested in why there is such a correlation. In addition, the author also pointed out many shortcomings of the regression model, which makes the results of this study seem immature.

7. Conclusions drawn in this study is apparent.

Reviewer #2: General Comment：

This manuscript explored the relationship between shale oil production and groundwater constituent levels in the Permian Basin. Produced water (PW) samples from active unconventional oil wells provided good reliability for the correlation relationship. The impact of shale gas/oil production on groundwater is an interesting topic. This manuscript also has a good review on the impact of shale oil/gas production on groundwater, but this manuscript did not discuss the long-term geochemical reaction of re-injected water in the reservoir. Valid conclusions are lacking and deserve to be discussed in depth. Thus, the reviewer can not recommend it for publication in the current form. A minor revision may be necessary. Following comments may be helpful:

Comment-1: The author pointed out the dramatic increase of the TDS, chloride, sodium, and calcium in the groundwater due to boom of shale gas/oil production. However, the mechanism of interaction between these elements is still unclear. Furthermore, what potential environmental hazards will be caused? Please explain it.

Comment-2: If the increase of the TDS, chloride, sodium, and calcium is an obvious trend, these related compounds play what’s role in geological formation? Please explain it.

Comment-3: The direction of Fig. 4 is not conducive to reading, it is recommended to retype.

Comment-4: The appendix files (Fig.S1 and Table S1) can not be found in the manuscript. Please check it.

Comment-5: The core of this manuscript is a linear regression formula (Eq.(1)), which lacks theoretical innovation. Please further explain the contribution of this manuscript.

Comment-6: It is recommended to merge the conclusion and discussion together.

6. PLOS authors have the option to publish the peer review history of their article (what does this mean?). If published, this will include your full peer review and any attached files.

Reviewer #1: No

Reviewer #2: No

---

## [Author Response · Author response to Decision Letter 0]

24 Mar 2021

Please see the attached Response to Reviewers file for detailed point-by-point responses.

---

## [Decision Letter · Decision Letter 1]

14 Apr 2021

Shale oil production and groundwater: What can we learn from produced water data?

PONE-D-21-02697R1

Dear Dr. Wang,

We’re pleased to inform you that your manuscript has been judged scientifically suitable for publication and will be formally accepted for publication once it meets all outstanding technical requirements.

Kind regards,

Jianguo Wang, PhD

Academic Editor

PLOS ONE

Additional Editor Comments (optional):

Reviewers' comments:

Reviewer's Responses to Questions

**Comments to the Author**

1. If the authors have adequately addressed your comments raised in a previous round of review and you feel that this manuscript is now acceptable for publication, you may indicate that here to bypass the “Comments to the Author” section, enter your conflict of interest statement in the “Confidential to Editor” section, and submit your "Accept" recommendation.

Reviewer #1: All comments have been addressed

Reviewer #2: All comments have been addressed

2. Is the manuscript technically sound, and do the data support the conclusions?

Reviewer #1: Yes

Reviewer #2: Partly

3. Has the statistical analysis been performed appropriately and rigorously? 

Reviewer #1: Yes

Reviewer #2: Yes

4. Have the authors made all data underlying the findings in their manuscript fully available?

Reviewer #1: Yes

Reviewer #2: Yes

5. Is the manuscript presented in an intelligible fashion and written in standard English?

Reviewer #1: Yes

Reviewer #2: Yes

6. Review Comments to the Author

Reviewer #1: The authors have adequately addressed my comments raised in a previous round of review and this manuscript is now acceptable for publication.

Reviewer #2: This is a review of the manuscript " Shale oil production and groundwater: What can we learn from produced water data? ". I appreciate the detailed responses of the authors. This manuscript has been greatly improved after the author’s revision. Thus, I thought it could be accepted to publication.

7. PLOS authors have the option to publish the peer review history of their article (what does this mean?). If published, this will include your full peer review and any attached files.

Reviewer #1: No

Reviewer #2: No

---

## [Editor Report · Acceptance letter]

22 Apr 2021

PONE-D-21-02697R1 

Shale oil production and groundwater: What can we learn from produced water data? 

Dear Dr. Wang:

I'm pleased to inform you that your manuscript has been deemed suitable for publication in PLOS ONE. Congratulations! Your manuscript is now with our production department. 

Kind regards, 

on behalf of

Dr. Jianguo Wang 

Academic Editor

PLOS ONE